# A Pilot Study for *Legionella pneumophila* Volatilome Characterization Using a Gas Sensor Array and GC/MS Techniques

**DOI:** 10.3390/s23031401

**Published:** 2023-01-26

**Authors:** Rosamaria Capuano, Antonella Mansi, Emilia Paba, Anna Maria Marcelloni, Alessandra Chiominto, Anna Rita Proietto, Andrea Gordiani, Alexandro Catini, Roberto Paolesse, Giovanna Tranfo, Corrado Di Natale

**Affiliations:** 1Department of Electronic Engineering, University of Rome Tor Vergata, Via del Politecnico 1, 00133 Rome, Italy; 2Interdepartmental Centre for Volatilomics ‘A. D’Amico’, University of Rome Tor Vergata, Via del Politecnico 1, 00133 Rome, Italy; 3Department of Occupational and Environmental Medicine, Epidemiology and Hygiene, Italian Workers’ Compensation Authority (INAIL), Via Fontana Candida 1, Monte Porzio Catone, 00078 Rome, Italy; 4Department of Chemical Science and Technology, University of Rome Tor Vergata, Via della Ricerca Scientifica, 00133 Rome, Italy

**Keywords:** gas sensor array, volatile organic compounds, *Legionella pneumophila*, volatilome

## Abstract

Legionellosis is a generic term describing the pneumonic (Legionnaires’ disease, LD) and non-pneumonic (Pontiac fever, PF) forms of infection with bacteria belonging to the genus *Legionella*. Currently, the techniques used to detect *Legionella* spp. in water samples have certain limitations and drawbacks, and thus, there is a need to identify new tools to carry out low-cost and rapid analysis. In this regard, several studies demonstrated that a volatolomics approach rapidly detects and discriminates different species of microorganisms via their volatile signature. In this paper, the volatile organic compounds (VOCs) pattern emitted in vitro by *Legionella pneumophila* cultures is characterized and compared to those produced by other *Legionella* species and by *Pseudomonas aeruginosa*, using a gas sensor array and gas chromatograph mass spectrometer (GC-MS). Bacterial cultures were measured at the 3rd and 7th day after the incubation. Sensor array data analyzed via the K-nearest neighbours (k-NN) algorithm showed a sensitivity to *Legionella pneumophila* identification at around 89%. On the other hand, GC-MS identified a bouquet of VOCs, mainly alcohols and ketones, that enable the differentiation of *Legionella pneumophila* in respect to other waterborne microorganisms.

## 1. Introduction

The term Legionellosis includes all forms of illnesses caused by Gram-negative aerobic bacteria belonging to the genus *Legionella* that occur in two main clinical manifestations of different severity [1]. Legionnaires’ disease (LD) is characterized by a pneumonia whose severity ranges from a mild cough to a rapidly fatal pneumonia. The death rate of Legionnaires’ disease depends on the severity of the pathology, and overall, it is usually within the range of 5–10%. In contrast to LD, Pontiac fever is an acute self-limiting and flu-like illness, and it does not affect the lung [2]. 

Although several of the more than 60 species described in the genus *Legionella* may cause disease [3], *L. pneumophila* serogroup 1 has been reported as the agent most frequently responsible for Legionellosis in Europe, where an increase in the number of cases has been recorded in recent years [4]. According to data from the European Centre for Disease Prevention and Control (ECDC) relative to 27 European countries, in 2020, a total of 8372 cases of Legionellosis were reported, of which 72% were reported by four countries with Italy in first place followed by France, Germany and Spain [5]. *Legionella* spp. are waterborne pathogens and they occur in natural freshwater habitats such as lakes, rivers and ground waters [6]. From natural sources, *Legionella* may enter into man-made water systems (potable water distribution, air-conditioning units, cooling towers etc.), where it can survive and proliferate [7]. The transmission from environmental sources to humans occurs mainly via inhalation of *Legionella*-contaminated aerosols [8]. In living and working environments, devices generating infectious aerosols such as showerheads, hot tubs, humidifiers, irrigation systems, etc. have been associated with the transmission of the microorganism to humans [9].

According to the provisions of the European Directive 2000/54/EC [10], all *Legionella* species are classified according to risk, group 2 being biological agents that can cause diseases in human subjects and be a hazard to workers. Among community-acquired pneumonia, the workplaces more frequently associated with Legionellosis cases are industrial settings, office buildings, healthcare facilities, ships and water and sewage plants [11]. Thus, the assessment of exposure risk to Legionella is essential in all living and working environments in order to implement appropriate prevention and control measures. In 2015, the Italian National Institute of Health, in compliance with the provisions of Legislative Decree n. 81/2008, which applies to all activities in which workers are potentially exposed to biological agents, issued “*Guidelines for the prevention and the control of the Legionellosis*” [1]. It provides technical guidance for the assessment and management of *Legionella* risk and describes the procedures for sampling and analysis of water samples. 

Currently, the main techniques used to detect *Legionella* spp. are the molecular methods (PCR or qPCR) and the culture method, but both methods have certain limitations and drawbacks [12]. The culture method has long been considered the gold standard for detecting *Legionella* from water samples though it has the disadvantage of being time-consuming [13]. Moreover, it presents high variability in enumeration [14], due to the inability to detect viable but not culturable (VBNC) cells, which are still alive but unable to grow on routine bacteriological media [15]. In contrast, the molecular methods have the advantage of being rapid (1 day), but the main drawback is represented by the fact that they are not able to distinguish between not viable and viable cells [16,17]. Thus, the necessity to develop new rapid and reliable methods emerges, in order to evaluate water quality and prevent the spread of new cases during Legionnaires' outbreaks. 

For this purpose, volatile organic compounds (VOCs) emitted by different microorganisms have been deeply investigated for several years, in order to identify microbial contaminations in different application contexts, including environmental monitoring. These compounds are the final products of metabolism of different microorganisms, such as fungi, molds or bacteria, and include different classes of organic chemicals. The VOCs profile is strictly correlated with microbial species and strains [18].

The gas chromatography linked to mass spectrometry (GC-MS) is the gold standard analytical technique for VOCs analysis, allowing the compounds emitted by the microbiological sample to be identified and quantified. However, this technique has high operating costs, and the analysis is time-consuming.

Interestingly, GC-MS studies point out that the same classes of chemicals and specific compounds could be shared by different strains of microorganisms; therefore, the key information for their identification rather than in individual VOCs can be found in the global VOC patterns. This is a typical situation where an array of cross-selective gas sensors can be successfully applied. In such an array, thanks to the partial selectivity of the sensors, a mixture of VOC is encoded in a pattern of sensor signals that, in some cases, preserve the relevant information necessary for the sample identification. The use of gas sensor arrays ensures a rapid, cheap and simple analysis of VOCs matrices. Sensor arrays have been successfully applied for the identification and differentiation of microorganisms in vitro [19,20] and also in environmental monitoring scenarios [21,22,23].

The final objective of this study was to define a volatile fingerprint associated with this bacterium, with the perspective of using these tools in the operational field, for environmental analysis in sites with a high risk of *L. pneumophila* contamination.

In this study, we analyzed the headspace of *L. pneumophila* broth culture, using an array of quartz crystal microbalance (QMB) gas sensors functionalized with molecular layers of porphyrinoids, in order to confirm the hypothesis that the volatolomics profile contains enough information to selectively identify *L. pneumophila.*

The selectivity was tested by comparing the volatilome of *L. pneumophila* with those released by a pool of other *Legionella* species and *Pseudomonas aeruginosa*. 

Sensor results were also validated with standard chromatographic analysis by means of SPME-GC/MS methods.

*Pseudomonas aeruginosa*, as *Legionella* spp, is classified in the second risk group (Italian Legislative Decree 81/2008). It is a Gram-negative bacterium and is present in all parts of the world in soil and water. It is the species most commonly associated with nosocomial infections following contact with contaminated fluids or surfaces. Often this pathogen is simultaneously present within *Legionella*-contaminated waters [24].

To the best of our knowledge, the literature is very limited about studies concerning *Legionella* culture VOCs. El Qader et al. analysed volatilome of bacteria and viruses responsible for pulmonary infections, and *L. pneumophila* was among the tested microorganisms [25]. 

## 2. Materials and Methods

### 2.1. Samples 

American Type Culture Collection (ATCC^®^) bacterial strains belonging to Legionella and Pseudomonas genera were used in this study. 

L. pneumophila (LP) strain Philadelphia-1 (ATCC^®^ 33152), L. dumofii (ATCC^®^ 33279) and L. anisa (ATCC^®^ 35292) strains were grown on buffered charcoal yeast extract (BCYE) agar medium supplemented with L-cysteine, ferric pyrophosphate and α-ketoglutarate (Oxoid Ltd., Basingstoke, UK) and bacterial colonies were macroscopically visible on plates after 4–5 days of incubation at 36 ± 1 °C. 

Pseudomonas aeruginosa (PA) strain ATCC^®^ 10145 was isolated in nutrient agar (NA) and incubated at 36 ± 1 °C.

Then, liquid cultures were prepared from single strains and transferred to 10 mL of BYEα broth (N-(2-acetamido)-2-aminoethanesulfonic acid-buffered yeast extract broth supplemented with 0.1% α-ketoglutarate, 0.04% L-cysteine and 0.025% iron III diphosphate, final pH, 6.9), followed by incubation at 36 ± 1 °C until reaching the exponential growth phase (OD_600_ of 0.6, corresponding to about 3 × 10^8^ CFU/ml) with constant shaking [26].

Lastly, 50 µL of each bacterial suspension was inoculated into 20 mL sterile vials crimped containing 5 mL of BYEα broth, and incubated at 36 ± 1 °C for 3 and 7 days. Only in the case of Legionella non-pneumophila species (L. dumofii and L. anisa), we inoculated 25 µL of each bacterial suspension into vials obtaining a single sample named Legionella spp. (LSpp).

On the 3rd day of incubation the mean value of bacterial concentration in inoculated vials was 1 × 10^6^ CFU/ml, while on the 7th day this value decreased slightly because of reduced availability of nutrients in the culture medium.

Not-inoculated BYEα broth (culture media, CM) vials were used as negative controls.

Experiments were carried out at the Biological Agents Risk Laboratory of the Department of Occupational and Environmental Medicine, Epidemiology and Hygiene, INAIL (Monteporzio Catone, Rome, Italy).

### 2.2. Gas Sensor Array Analysis

For analysis of culture headspace, a gas sensor array designed and developed by the University of Rome Tor Vergata was used. 

The system core consists of a matrix of ten quartz microbalance (QMB)-based gas sensors. For these sensors, slight mass changes (Δm) in the absorbing layer of the quartz surface result in a frequency shift (Δf) of the electrical output signal of the oscillator circuit. In a regime of small perturbations, Δf is linearly proportional to Δm [27]. The adopted QMBs are AT-cut quartz, having a fundamental frequency of 20 MHz, corresponding to a mass resolution of the order of a few nanograms [28]. 

Each QMB is functionalized via spray casting using different coating materials (porphyrins and corroles). Seven of them are coated with six metal complexes (Mg, Co, Cu, Zn, Mn, Sn) and a free base (H_2_) of 5,10,15,20-tetrakis-(4-butyloxyphenyl)porphyrin (TBPP); the other three sensors were functionalized by free base (H_3_) and phosphorus and manganese complexes of 5,10,15-triphenylcorrole (TPC) [29].

Sensing molecules were synthesized at the laboratories of the Department of Chemical Science and Technology, University of Rome Tor Vergata, and fully characterized in terms of VOCs sensitivity in previous works [30,31,32,33].

QMBs are singularly connected to an oscillator circuit. Temperature-compensated quartz is the reference for measuring the frequencies of the oscillator outputs, allowing for a frequency resolution of 0.1 Hz. The system is also equipped with temperature and relative humidity sensors. The gas delivery is managed through a tubeless embedded pneumatic system comprising a PMMA (poly(methyl methacrylate)) manifold, provided by two inlets and one output, where a miniature diaphragm pump (flow range: 0–200 sccm), one three-way electron valve, a proportional electron valve and a flow sensor are connected. 

The instrument is connected and powered via a USB connection. Data acquisition, function and settings of the instrument are managed via in-house software in Matlab.

Furthermore, 20 mL headspace glass vials (VWR International Srl, Milano, Italy) containing 5 mL bacterial cultures are maintained at a constant temperature of 37 °C in a thermostatic water bath in order to equilibrate headspace composition during measurements.

Volatile organic compounds emitted from different samples were analyzed for 40 s via a sensor array with a flow rate of 100 sccm (standard cubic centimeters per minute). A filter with a porosity of 0.2 µm (Sartorius, Germany) was used during the analysis in order to avoid the accidental migration of bacteria from the samples to the device. The stable sensors baseline was obtained by flowing the laboratory air filtered through a CaCl_2_ bed. Measurements were carried out under a biohazard hood. Figure 1 shows the experimental setup.

All bacterial cultures were analysed after 3 and 7 days of incubation and each condition was produced in triplicate for the different strains. Each sample was measured three times.

As negative control, measurements of BYEα broth headspace, incubated at days 3 and 7, were also performed (Figure 1).

Figure 2 shows the experimental outline of gas sensors analysis.

A total of 36 samples were analyzed:Three samples of BYEα broth incubated for 3 days and three samples incubated for 7 days;Three samples of *L. species* and *P. aeruginosa* incubated for 3 days and 7 days;A total of 18 samples of *L. pneumophila*: 9 incubated for 3 days and 9 incubated for 7 days.

### 2.3. GC-MS Characterization

Bacterial cultures were inoculated and incubated for 7 days in glass vials crimped with caps having a viton/PTFE septum (ChromSelection, distributed by HTA s.r.l., Brescia, Italy), and 3 biological replicas for each considered bacteria were prepared and analyzed (Figure 3).

Volatile organic compounds emitters using different microorganisms were preconcentrated using a solid phase micro-extraction (SPME) technique. Using an autosampler (HT2800T, HTA, Brescia, Italy), 50/30 μm Divinylbenzene/Carboxen/PDMS (SUPELCO, Bellefonte, PA, USA) fiber were immersed in culture headspace for 1 h, maintaining the sample at a constant temperature of 40 °C during the extraction time (Figure 1). 

Analyses were performed at the Interdepartmental Centre for Volatilomics ‘A. D’Amico’ (CIVAD-Univ. Rome “Tor Vergata”) with a GC-MS (Shimadzu GCMS-QP2020 NX, Kyoto, Japan) equipped with a SH-I-5MS capillary column (30 m length × 0.25 mm I.D. × 0.25 μm thickness, Shimadzu, Kyoto, Japan). This is a general purpose non-polar fused silica column, having a cross-linked 5% diphenyl/95% dimethyl polysiloxane phase. 

The VOCs adsorbed in the SPME were desorbed from the fiber at 250 °C for 3 min in the GC injection port, operating in splitless injection mode. Compound separation took place on the GC column using the following oven conditions: initial temperature of 40 °C held for 5 min, then increased by 7 °C/min to reach 220 °C and ramped by 15 °C/min up to 300 °C, which was held for 3 min (total runtime: 39 min). Ultra-high purity helium was used as carrier gas, working in linear velocity constant mode, maintaining linear velocity at 30.2 cm/s.

The mass spectrometer was a single quadrupole analyzer in electron ionization mode (energy 70 eV) and was set to record between *m/z* 40 and 450  in the full scan mode. The temperature of the transfer line and ion source was 250 °C. GC-MS data were analyzed using the section GCMS post-run analysis of the GCMS solutions software (version 4.52, Shimadzu Corporation, Kyoto, Japan).

Putative identification of compounds was performed by comparing target peak mass spectra with those present in the NIST20 library. 

### 2.4. Fast GC/MS

The VOCs, emitted by the various bacterial cultures (Figure 3) placed in headspace vials after incubation, were pre-concentrated using the solid phase microextraction (SPME) technique. Measurements were performed at the Department of Occupational and Environmental Medicine, Epidemiology and Hygiene (INAIL), in parallel with those conducted via conventional GC-MS at CIVAD (Figure 1 and Figure 3). A Perkin Elmer Custodion SPME 18G BT syringe was used, whose fiber was composed of Divinylbenzene/Polydimethylsiloxane (DVB/PDMS), which was immersed in the culture headspace for 1 h, keeping the sample at the constant temperature of 40 °C.

The instrument used to analyze the organic compounds absorbed on the fiber was a portable GC/MS, Torion^®^ T-9 (Perkin Elmer, Waltham, MA, USA) (Figure 4).

The Torion^®^ T-9 is a portable GC coupled to a toroidal ion trap mass spectrometer (TMS) with a mass range from 41 to 500 Daltons, equipped with an MXT-5 (5 m length × 0.1 mm I.D. × 0.4 μm thickness) metal capillary column. This is a low polarity column consisting of a fixed phase of 5% diphenyl and 95% dimethyl polysiloxane. The chromatographic conditions used are the following: 10:1 split injection mode; injector temperature 270 °C; He as gas transport; the programmed oven temperature ramp starts from 50 °C, where it stays for ten s, and then increases by 2 °C/s until it reaches 290 °C, where it stays for 60 s. The run has a total duration of 190 s.

The VOCs captured by the SPME were desorbed from the fiber at 270 °C for 20 s, directly in the injection port of the GC. The mass spectrometer toroidal ion trap is set in the electron ionization mode (energy 70 eV) recording data in full scan mode, between *m/z* 41 and 500 Daltons. The data obtained were analyzed using Chromion software (Perkin Elmer version 2.0.4.0), and the identification of the compounds was performed by comparing the mass spectra of the detected peaks with those present in the NIST Mass Spectral Search Program 2017 (Version 2.3).

### 2.5. Sensor Data Analysis

Sensor responses consist of a 10-dimensional vector, containing the variation in resonant frequency measured at the beginning and at the end of the exposure to the sample. 

Sensor data have been classified using the K-nearest neighbours algorithm. All calculations were performed in Matlab R2020b, implemented using Statistics and Machine Learning Toolbox (Mathworks, Natick, MA, USA).

## 3. Results and Discussion

### 3.1. Gas Sensor Array 

Each kind of sample was produced in triplicate in order to account for biological variability between cultures. 

Three consecutive measurements of the headspace were performed on each bacterial culture.

Figure 5 shows the boxplot related to the distribution of sensor responses at different classes of bacteria and to BYE broth (CM), considering data obtained via samples incubated at 3 and 7 days.

From those plots it is possible to see how the analysis of the headspace of the BYEα-broth gives rise to statistically lower frequency variations, compared to the measurements of the bacterial cultures. This suggests that the headspace of uninoculated medium is less rich in volatile organic compounds, and they did not influence the sensors’ response to different bacterial strains. There is no significant difference between sensor signals obtained from each bacterial species incubated for 3 days compared to those incubated for 7 days. Furthermore, from the responses of the single sensor it is not possible to discriminate between *L. pneumophila* and other bacterial cultures.

In order to evaluate the suitability of the gas sensor array to recognize the volatile fingerprint related to LP, data patterns were analyzed via the classification multivariate approach using the K-nearest neighbours algorithm, considering Euclidean distance and k = 1 [34].

Based on the evidence discussed above, no distinction was considered with respect to the incubation time of the analyzed samples.

The classification model was calculated considering the entire data-set of measures, collected during three considered days, and was cross-validated via leave-one-out. Data were previously autoscaled (having zero mean and unitary variance), in order to remove any influence due to concentration [35].

The confusion matrix in Table 1 shows the obtained classification results.

The obtained model accuracy is around 87% with a sensitivity and a precision with respect to *L. pneumophila* of around 89%.

The sensitivity and precision are defined as follows:(1)$$\mathrm{Sensitivity}=\frac{\mathrm{True positive samples}}{\mathrm{Real positive samples}}$$
(2)$$\mathrm{Precision}=\frac{\mathrm{True positive samples}}{\mathrm{true positive}+\mathrm{false positive}}$$

BYEα broth (CM) is completely discriminated with respect to bacterial cultures.

A good sensitivity is also obtained for the discrimination of Legionella species and P. aeruginosa, around 78%. 

The obtained results show a good suitability of the system to recognize *L. pneumophila* compared to other bacterial cultures.

### 3.2. Gas Chromatography-Mass Spectrometry (GC-MS)

To carry out the analysis of the data obtained through measurements with GC-MS, the chromatogram relating to the total ion current (TIC) was considered. Chromatographic peaks were automatically integrated and the abundance of each compound (area under peak curve) was considered for data analysis.

Figure 6 shows the characteristic chromatograms obtained from measurements of BYEα broth, *L. pneumophila*, *L. species* and *P. aeruginosa*.

The headspace-SPME technique coupled with GC/MS analysis allowed a total of 37 compounds to be extracted.

For the analysis, only those compounds present in at least 2 of the 3 samples of the single species and identified with more than 70% of similarity, with respect to the target mass spectrum, were considered. In general, the headspace of BYE🡐α broth samples contains less VOCs compared to bacterial cultures.

With respect to what we found, we can divide compounds into three categories: those common in more samples and not statistically significant, those characteristics of one of the bacteria species and compounds common to the three bacterial samples and statistically different between them, as it is possible to appreciate in Figure 7.

In this bar plot, each colored band corresponds to a compound with its respective abundance and bands are sorted with respect to VOC elution time.

The boxplot in Figure 8 shows the abundance distribution of the 17 most significative compounds.

It is immediately clear that there are evident differences between VOCs emitted by the different bacterial species, both from a qualitative and quantitative point of view.

Three compounds are present only in the headspace of *L. pneumophila*: pentanol, methyl isovalerate and 2,7-dimethyl-4,5-octanediol. Methyl isovalerate is a short chain fatty acid produced by other kinds of bacteria such as *Streptococcus mutans*, *Bacillus atrophaeus* GBSC56 [36,37]. It is interesting to note that these compounds are not found in the headspace of other *Legionella* species. 

3-methyl-1-butanol and 3-methyl-1-butanol acetate are quantitative higher in LP compared to the other bacterial cultures. The first compound has been reported by Qader et al. as present in *L. pneumophila* culture and associated bacterial activity [25].

*P. aeruginosa* is characterized by production of long chain VOCs such as 2-nonanone, 2-nonanol, 2-butil-1-octanol, 2-undecanone and 2-undecanol, sulphur containing volatile compounds. In the literature it is reported that 2-nonanone correlates directly with the proliferation rate of *P. aeruginosa* [31]. 3-methyl-1-butanol, 2,5-dimethylpyrazine and volatile sulphur-containing compounds 2-butil-1-octanol, 2-undecanone and 2-undecanol are common volatile organic compounds found in *in vitro* and *in vivo* studies of samples with PA [38,39,40,41]. 

### 3.3. Fast GC/MS

The comparison of the chromatographic peaks, obtained by analyzing the cultures headspace via GC/MS, with the NIST library data, provided the identification of different organic compounds patterns for the three different species.

*L. pneumophila* emitted a mixture of VOCs, in which the most intense peaks are generated by 1 butanol-3 methyl (C5H12O, molecular weight 88.1482, CAS Registry Number 123-51-3) and methyl isovalerate (C6H12O2, molecular weight 116.1583, CAS Registry Number: 556-24-1).

The chromatogram is reported in Figure 9.

Regarding *L. species*, the emission of volatile organic compounds seems to be lower both in number and intensity of peaks, but the main substance detected was identified as 1 butanol-3 methyl, the same as for *L. pneumophila*. The chromatogram is reported in Figure 10.

The analysis of the headspace of the culture of *P. aeruginosa* presents a completely different chromatographic profile, producing a series of volatile substances, the most abundant of which was identified as 2-nonanone (C9H18O, molecular weight 142.2386, CAS Registry Number 821-55-6). Other substances identified are 2-undecanol (C11H24O, molecular weight 172.3077, CAS Registry Number 1653-30-1), 2-undecanone (C11H22O, molecular weight 170.2918, CAS Registry Number 112-12-9) and dimethyl sulfone (C2H6O2S, molecular weight 94.133, CAS Registry Number 67-71-0). The relative chromatogram is reported in Figure 11.

As we can see, although identified compounds are numerically lower than those obtained with the common GC-MS, the obtained results are completely comparable. 

## 4. Conclusions

In this pilot study, the capability of a gas sensor array for *in vitro* discrimination of *L. pneumophila* was investigated. In order to evaluate the discriminative system performances, headspace of *L. pneumophila*, other *L*. species and *P. aeruginosa* were analyzed with the gas sensor array. As reference, the volatilome of not-inoculated culture media was considered.

Sensor results show that the device is able to discriminate totally between not-inoculated and inoculated BYEα broth (culture media).

Moreover, *L. pneumophila* was recognized with 89% sensitivity with respect to other samples.

GC-MS analysis performed on considered cultures showed characteristic VOC patterns for *L. pneumophila*, *P. aerugionosa* and other *L.* species, confirming data reported in the literature, from previous studies, and supporting the results obtained with the sensor system. 

To the best of our knowledge, not many works are present in literature concerning the characterization of *L. pneumophila* VOC profile. In the present study, the presence of pentanol, methyl isovalerate and 2,7-dimethyl-4,5-octanediol in culture headspace fully characterizes *L. pneumophila* compared to other considered bacteria.

The obtained results, although preliminary, lay the foundations for further investigations on environmental samples. Such studies will require future investigation that includes the evaluation of the ability of electronic noses to detect *L. pneumophila* in real water samples collected from different sources (cooling towers, water tanks, showerheads, etc.) and also in respect to other microorganisms that simultaneously could be present within the *Legionella*-contaminated waters.

The final aim of this research is to develop a rapid, low-cost and portable system capable of detecting *L. pneumophila* in water samples to overcome the limitation of currently used detection methods. 

## Figures and Tables

**Figure 1 sensors-23-01401-f001:**
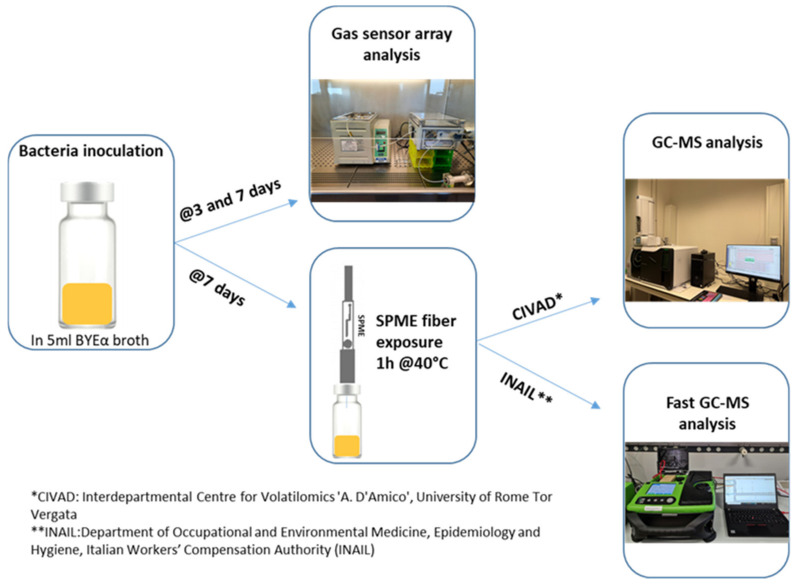
Schematic representation of experimental workflow.

**Figure 2 sensors-23-01401-f002:**
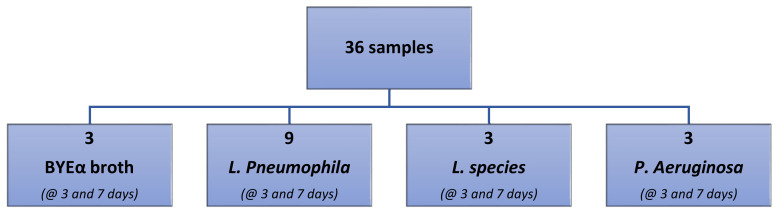
Experimental outline of gas sensor array analysis.

**Figure 3 sensors-23-01401-f003:**
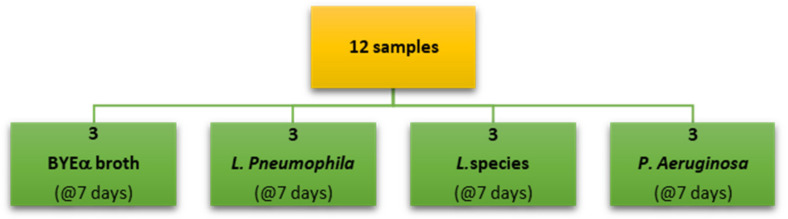
Experimental outline of GC-MS and fast GC-MS analysis.

**Figure 4 sensors-23-01401-f004:**
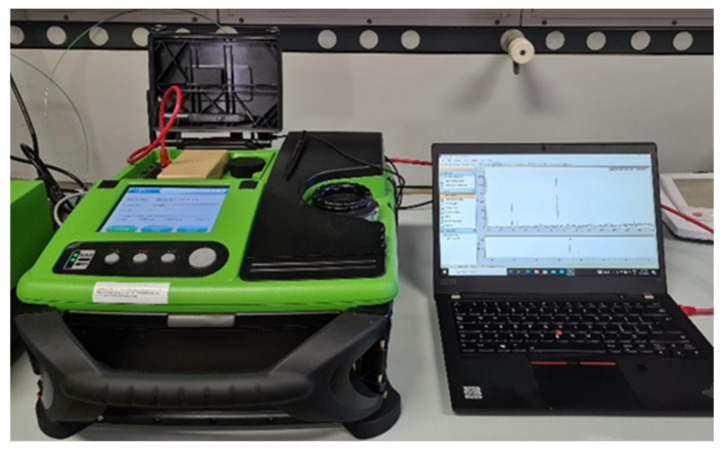
The portable GC-MS Torion^®^ T-9 interfaced with its computer.

**Figure 5 sensors-23-01401-f005:**
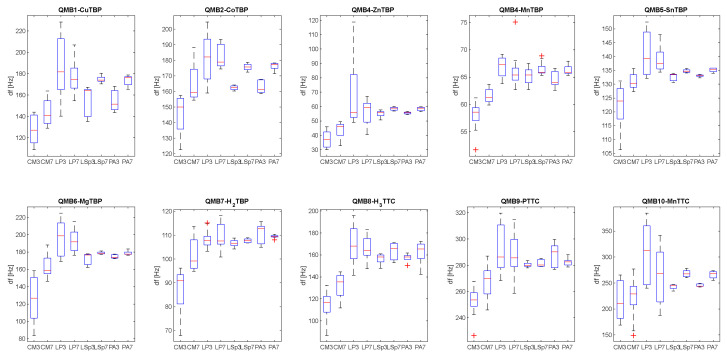
Boxplot representing the distribution of the sensor responses to control media (CM) and bacteria cultures: *L. pneumophila* (LP), *Legionella* species (LSp) and *Pseudomonas aeruginosa* (PA) considered after 3 (e.g., LP3) and 7 (e.g., LP7) days of incubation.

**Figure 6 sensors-23-01401-f006:**
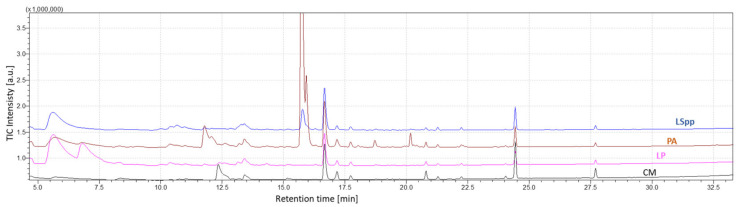
Comparison of chromatograms obtained from analysis of culture headspace of *Legionella* species (blue line), *Pseudomonas aeruginosa* (red line), *Legionella pneumophila* (pink line) and not-inoculated BYEα broth (CM-black line). Samples were inoculated for 7 days before the analysis.

**Figure 7 sensors-23-01401-f007:**
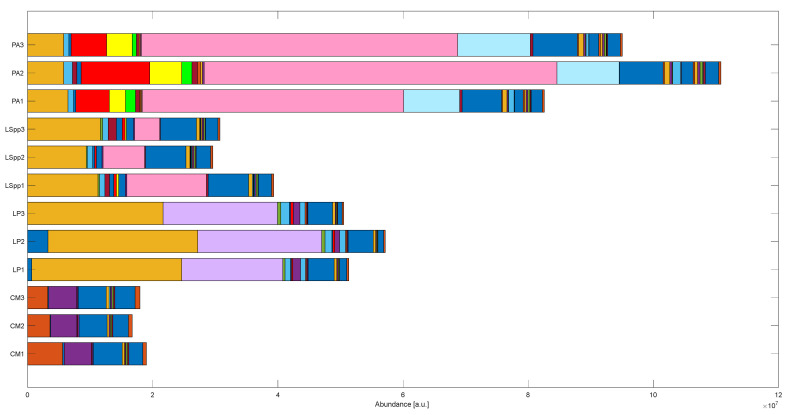
Bar plot representing compound pattern of each analyzed sample of BYEα broth (CM), *L. pneumophila* (LP), *L. species* (LSpp) and *P. aeruginosa* (PA) incubated for 7 days. Each colored bar represents the compound abundance and bands are sorted with respect to VOC retention time.

**Figure 8 sensors-23-01401-f008:**
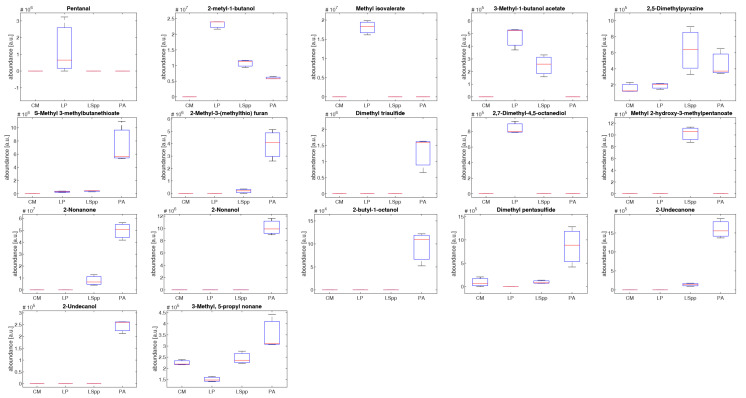
Boxplot representing the distribution of different characteristic compound abundances for the different kinds of cultures: BYEα broth (CM), *L. pneumophila* (LP), *L.* species (LSpp) and *P. aeruginosa* (PA).

**Figure 9 sensors-23-01401-f009:**
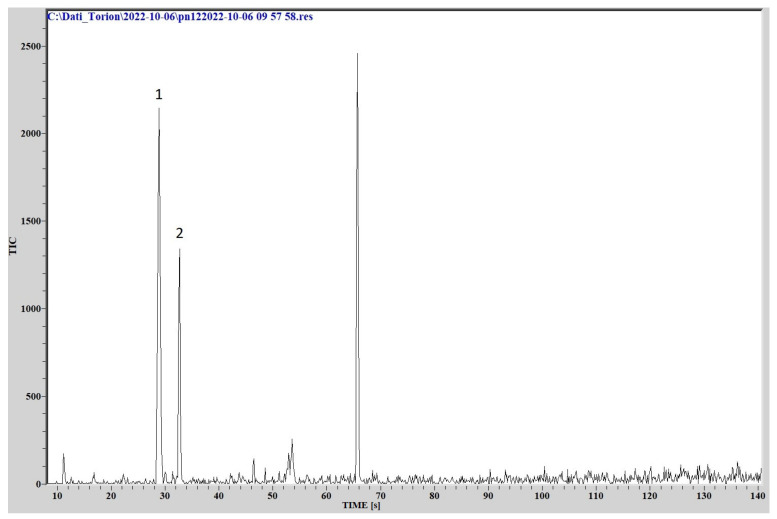
Chromatogram from *L. pneumophila*. (1) 1-butanol- 3methyl, (2) Methyl isovalerate.

**Figure 10 sensors-23-01401-f010:**
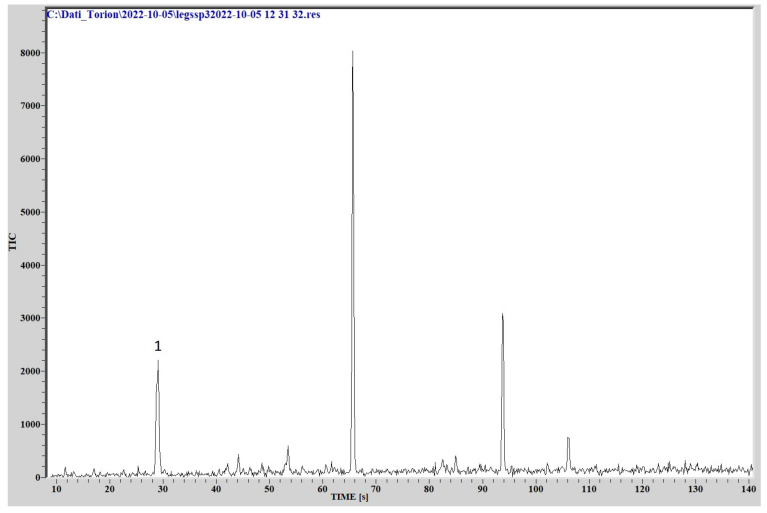
Chromatogram from *L.* species. (1) 1butanol- 3methyl.

**Figure 11 sensors-23-01401-f011:**
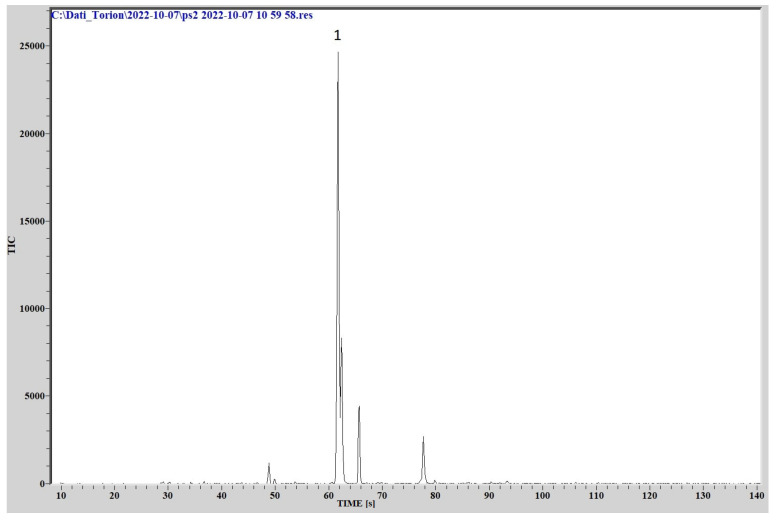
Chromatogram for *P. aeruginosa.* (1) 2-nonanone.

**Table 1 sensors-23-01401-t001:** Confusion matrix obtained via k-NN classification model calculated using gas sensor array data, considering Euclidean distance and k = 1. The elements on the diagonal correspond to correctly classified measurements.

Actual Class	Predicted Class				
		**CN**	**LP**	**LSpp**	**PA**
	CN	21	0	0	0
	LP	0	48	2	4
	LSpp	0	3	14	1
	PA	0	3	1	14

## Data Availability

All data that support the findings of this study are available after the reasonable request to the corresponding author.

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
