# Peer review of "A Pilot Study for *Legionella pneumophila* Volatilome Characterization Using a Gas Sensor Array and GC/MS Techniques"

_sensors, 2023, doi:10.3390/s23031401_

Round 1

Reviewer 1 Report

Review report

A pilot study for Legionella pneumophila volatilome characterization by a gas sensor array and GC/MS techniques.

Brief summary

From this manuscript, Rosamaria Capuano and colleagues developed the protocol to investigate volatile organic compounds (VOCs) patterns emitted by Legionella pneumophila using a gas sensor array and GC/MS techniques. The authors show the characteristic compounds in order to differentiate from other similar microorganisms.

General comments

The manuscript was well organized and in order. It is easy to follow and understand. However, some information can be omitted such as the introduction part, and some are redundant. Minor English language modification is recommended including scientific forms (italic letter of bacteria species). Word choice in some sentences should be reconsidered.

Specific comments

Abstract

I suggest to abbreviate the introduction in the abstract because it is too long. The objection of this study should be emphasized as well as the new finding or developed protocol.

Introduction

1.     The first two paragraphs should be shortened especially the general information of infectious disease caused by Legionella pneumophila.

2.     I suggest to highlight how this tested bacteria leads to morbidity and mortality in human.

3.     Please clearly state the objective of this study and how to achieve it at the end of this part.

Methods

I highly suggest to draw or present the infographic of the methods in step by step instead of photographs because it is the experimental protocol.

Result and Discussion

This part is cleary described.

Conclusion

1.     I suggest to abbreviate the first paragraph since it is not necessary.

2.     Please mention that this is the pilot study. Why did the author present of only this preliminary result?

3.     What was the limitation?

4.     What is the future plan or recommendation?

5.     How is the significance of this study to the clinical field, environmental and application?

Reviewer 2 Report

The work is devoted to the important task - the first stage of development of methods for determination of Legionella pneumophila by volatile compounds using an array of sensors.  Namely, the possibility of identifying Legionella pneumophila using an array of sensors among other bacteria is checked, as well as the determination of volatile markers of Legionella by the gas chromatography with mass spectrometry.

 However, there are several questions for the authors that require clarification:

 1. In the section Materials and methods: an array of sensors is given, but no characteristics of the obtained sensors are indicated: sensitivity to classes of volatile compounds, film weight, film stability, etc. If some characteristics are defined earlier, then appropriate references can be given.

 2. On page 4, lines 173-177, check the number of claimed sensors and coatings.  For example, seven metal complex, although only six are listed.  And the origin of these metal complexes is not indicated (where they were bought or synthesized).

 3. In the results of paragraph 3.1, it would be logical to indicate which concentrations of microorganisms are detected by an array of sensors.  That is, after 3 days, how many CFU distinguish the sensors from a agar medium?

 4. Figures 9 and 11 are not very clear, it can be improved the clarity and sharpness so that the numbers on the axes are visible.  Figure 10 does not show the signature of peak 1.

 5. Line 182 abbreviation PMMA is not mentioned.

 6. In conclusion, it would be better to summarize the results in accordance with the objectives of the work, for example, to describe the typical volatile compouds for Legionella pneumophila. It would also be logical in the conclusion to add recommendations on the application of the results obtained in the analysis of real objects, the development of this research.
